# “Intervention Program Based on Self”: A Proposal for Improving the Addiction Prevention Program “Unplugged” through Self-Concept

**DOI:** 10.3390/ijerph19158994

**Published:** 2022-07-24

**Authors:** Cédrine Bourduge, Georges Brousse, Florence Morel, Bruno Pereira, Céline Lambert, Marie Izaute, Frédérique Teissedre

**Affiliations:** 1Université Clermont Auvergne, LAPSCO, CNRS, Institut Pascal, F-63000 Clermont-Ferrand, France; marie.izaute@uca.fr (M.I.); frederique.teissedre@uca.fr (F.T.); 2Université Clermont Auvergne, Clermont Auvergne INP, CHU Clermont-Ferrand, CNRS, Institut Pascal, F-63000 Clermont-Ferrand, France; 3Centre Hospitalier Universitaire de Clermont-Ferrand, Service d’addictologie et Pathologies Duelles, F-63000 Clermont-Ferrand, France; f_morel@chu-clermontferrand.fr; 4CHU Clermont-Ferrand, Unité de Biostatistiques (DRCI), F-63003 Clermont-Ferrand, France; bpereira@chu-clermontferrand.fr (B.P.); clambert@chu-clermontferrand.fr (C.L.)

**Keywords:** self-concept, prevention, unplugged, adolescents, gender

## Abstract

The “Intervention Program based on Self” (IPSELF) project was created to address the gap between the acquisition of life skills during prevention programs and their application with a session for developing one’s self-concept included in the European program “Unplugged”. The present study evaluated its effectiveness. A total of 157 middle school students (94 girls, 63 boys, *M_AGE_
*= 12.89, *SD* = 0.45) from three schools in France participated in this study. The participants attended one of two programs (Unplugged/IPSELF). The effectiveness of the IPSELF add-on session was measured with the Self-Concept Clarity Scale, and the differences between the two programs was measured with the prototype willingness model. Adolescents in IPSELF rated the typical nonsmoker and cannabis nonsmoker more favorably, and the typical drinker less favorably. They felt more different from the typical smoker and drinker after participation in IPSELF. More alcohol experimenters were observed in Unplugged. The knowledge gained in IPSELF appeared to help adolescents more than that gained in Unplugged to change their smoking behavior. Furthermore, IPSELF had a more beneficial effect for girls, who felt that they had gained more control over their alcohol and cannabis use than boys, whereas Unplugged had a more positive effect on boys, who gained better control over their consumption. Moreover, the girls felt that they had gained more knowledge about the substances discussed in IPSELF than in Unplugged. We therefore recommend the use of IPSELF especially with female audiences.

## 1. Introduction

In France, addictive behaviors are a major public health problem. French adolescents are the most frequent users of cannabis in Europe, and the second most frequent users of other illicit substances [1]. The use of tobacco, alcohol and cannabis has serious physical, mental and social consequences [2,3]. It is during adolescence that addictive behaviors begin and become settled, with more early and considerable use in boys than in girls [4]. Primary prevention allows for acting with adolescents before the first experiments. Thus, preventing addictive behaviors at an early age can delay the entry into drug use and thus limit the risk of addiction in adulthood [5]. Nevertheless, the literature has shown that girls are more positive about school-based primary prevention programs than boys [6]. It is therefore important to implement this type of action while taking into account gender differences during its evaluation [7].

For a long time, primary prevention programs focused on imparting knowledge about illicit substances, but these have had no significant effect [8]. Today, prevention programs are centered around the acquisition of life skills. For the World Health Organization, life skills consist in the ability of a person to respond effectively to the demands and trials of daily life [9]. The development of these skills during adolescence can prevent substance use [10], which accounts for their integration into more modern prevention programs. The program named “Unplugged” is one of these new programs. Unplugged is an interactive European school-based curriculum against youth substance use, based on the comprehensive social influence approach [11] and life skills development [12]. It has been developed, implemented and tested in seven European Union countries [13]. Its effectiveness in preventing or reducing daily tobacco use, heavy alcohol use, recent cannabis use [14], as well as binge drinking and tobacco and alcohol use [15,16], has made Unplugged an effective program for dealing with substance use in adolescents.

Despite the effectiveness of these modern prevention programs, some studies have highlighted a gap between the acquisition of life skills and their application in context by the adolescents [17]. The acquisition of life skills is not always put into practice in the refusal to use illicit substances in the face of strong peer pressure. It is in response to this finding that we propose to modify the Unplugged program by adding a specific session: the “Intervention Program based on Self” project (IPSELF). Through this add-on session for the Unplugged program, it is proposed to bridge the gap with a 13th session for developing one’s self-concept. One’s self-concept refers to the extent to which one’s self-knowledge is defined, consistent and currently applicable to one’s attitudes and dispositions [18]. A positive self-concept favors increased effort, perseverance in the face of difficulties, the use of acquired abilities and strategies, and increased efficiency [19]. It also allows for better academic performance [20] by facilitating the acquisition of knowledge. Moreover, knowing oneself well can reduce the impact of social influence on substance use [21]. Thus, incorporating the development of positive self-knowledge through adaptive and protective identity development into substance abuse prevention interventions could have a lasting and protective impact [22,23]. The proposal of this study is therefore to develop self-concept abilities. Specifically, our session focuses on self-definition through positive character traits, personal values and the possible impact of one’s self-definition [24], in order to make the self-concept more positive. To do this, the session offers adolescents several activities. The first activity is a brainstorming session asking adolescents about what defines them. This discussion shows them that a multitude of things can define us. The goal is also to arrive at the concept of value. In the second activity, adolescents are asked to choose a person they admire (real or fictional) and to indicate the values that this person represents. The values proposed as examples are all positive. They then choose which of the values they would like to improve in themselves. This exercise allows adolescents to put into words what is important to them, that is, their values [25]. In the final activity, the adolescents pair up. Each must choose two values that define their partner and give an explanation for their choice. This exercise helps to show adolescents that others may see them differently than they see themselves [24]. From this session, small exercises on self-concept are to be completed at home by the adolescents for each of the following sessions.

The objective of this study was to implement and evaluate the effectiveness of the “Intervention Program based on Self” program, consisting of the Unplugged program sessions and a complementary session on self-concept development, by comparing it to the reference program Unplugged.

In order to examine the impact of self-concept on the effectiveness of the Unplugged program, the present study relies on the prototype willingness model. The prototype willingness model focuses on reactive decision making and deliberative decision making in order to predict behaviors [26,27]. It proposes that behavior is co-determined by behavioral intentions and behavioral willingness [28]. Behavioral intentions are deliberatively formed plans of action that are derived on the basis of individuals’ attitudes and subjective norms, whereas behavioral willingness is a general openness to behave that increases the likelihood of a behavior when an individual encounters “facilitating situations” [26]. In the theory of planned behavior, behavioral intention is determined by attitude, perceived norm and perceived control [29,30]. The attitude consists in all beliefs about the likely consequences of the behavior. The perceived norm refers to beliefs about the normative expectations of others. Perceived control corresponds to beliefs about the presence of factors that may facilitate or impede the performance of the behavior. The more favorable the attitude and subjective norm, and the greater the perceived control, and the stronger the person’s intention to perform the behavior should be. This theory has often been used in the prediction of substance use in adolescents [31,32,33], but it does not include the Self; the development of the self-concept should thus not act on this dimension of the model. The model also proposes that behavioral willingness is determined by prototype perceptions. Prototype perceptions are positive or negative valences that refer to the cognitive representations that someone holds for the typical members of social categories. Two types of prototype perceptions are typically studied in this model: the prototype favorability perceptions (the positive or negative evaluation of the prototype) and the prototype similarity perceptions (how similar individuals believe themselves to be to the prototype). They are recognized as influencing behavior through reactive decision making (behavioral willingness) rather than deliberative decision making (behavioral intention): the more favorably their image is assessed, the more willing they are to engage in the behavior. This finding has been observed for smoking and drinking in many studies [34,35,36,37,38,39,40]. In addition, studies have also shown that the more similar adolescents’ smoking or drinking images are to their self-images, the greater their willingness to smoke or drink [41,42]. Changing adolescents’ self-concept could therefore alter their similarity to the consumer prototype and their willingness to consume harmful substances.

This is why, in the test of the “Intervention Program based on Self”, we hypothesize a greater clarity of self-concept and greater gains in knowledge about themselves, with greater change in self-perception, for adolescents in the IPSELF program compared to those in the Unplugged program after participation. We believe that this development of self-concept will allow adolescents to feel less similar to prototype substance users and more similar to non-users, which should decrease their willingness to consume harmful substances, and thus reduce their consumption of them. We also believe that the knowledge gained about harmful substances in IPSELF will be more helpful in changing use behavior in adolescents than that gained in Unplugged. However, we expect no impact of self-concept on attitude, norm or control towards substance use, and thus no impact on the intention to use. We will consider the effect of gender in order to observe the effectiveness of each program for girls and boys, due to the differences in consumption and prevention sensitivity identified in the literature.

## 2. Materials and Methods

### 2.1. Participants

One hundred and fifty-seven middle school students (94 (59.87%) girls and 63 (40.13%) boys) in grade 8 (aged from 11 to 14 years, *M* = 12.89, *SD* = 0.45) from three middle schools in the Auvergne Rhône Alpes region of France took part in this study. The classes in each middle school were randomly assigned (by lottery) to the two programs (Unplugged (*n* = 83 (52.87%) and IPSELF (*n* = 74 (47.13%)), ensuring that each school had the same number of classes for each program. Thus, the Unplugged and IPSELF programs were delivered at each middle school to an equivalent number of students to avoid possible differences between the middle schools (see Figure 1).

The middle schools were recruited on a voluntary basis. Students could only participate with parental consent. This study was conducted in accordance with ethical standards and received approval from local ethics committees (INSERM agreement reference: 19||134-00, ANSM registration number: 2019-A03131-56). The current study is available in the OSF repository at: https://osf.io/3fq6d/?view_only=4e0760af24f04a87aad67dc4267280e3 (accessed on 25 May 2022).

### 2.2. Procedure

The participants attended one of two prevention programs (Unplugged or IPSELF) at their middle school during the 2020–2021 school year. The distribution was conducted by lottery. Unplugged consists of 12 sessions and IPSELF of 13 (the 12 Unplugged sessions and an additional session developing self-concept). The sessions took place for one hour every two weeks in a half-group format. Because the IPSELF program had an additional session, we added a control session for the groups in the Unplugged condition (a game presented as a session working on communication) to avoid a bias related to the number of sessions. Session 10 of the program (“Drugs: Getting Informed”) had to be conducted virtually (via the Zoom software) during the lockdown period due to the COVID-19 pandemic. This change in format was common to both groups and the content was the same (acquiring knowledge about drugs). We know that the most important element in effective prevention is the acquisition not so much of knowledge as of life skills [8]. Thus, we do not believe that this change in format had a significant impact on the program’s effectiveness. In addition, using the digital format for this session allowed us to complete both programs in all groups.

A questionnaire was completed by the participants at the beginning (T1) (between 7 and 9 October 2020) and end (T2) of the program (between 16 June and 1 July 2021), in a digital format at their middle school, on a tablet via the Qualtrics online questionnaire creation software.

### 2.3. Materials

The measures in this study were based on the self-concept and the Prototype Willingness Model [26]. The different scales were presented in a random order to avoid bias. The description of the different measures and their Cronbach’s alpha (α) values are presented below.

#### 2.3.1. Self-Concept

*Self-concept clarity (SCC)* was measured by averaging a 12-item scale [43] (α_T1_ = 0.85, α_T2_ = 0.89). For each item, the participants indicated their level of agreement from 1 (agree) to 7 (disagree). The higher the score, the clearer the self-concept.

#### 2.3.2. The Prototype Willingness Model

*Prototype favorability* was measured for the typical smoker (α_T1_ = 0.60, α_T2_ = 0.71), the typical nonsmoker (α_T1_ = 0.71, α_T2_ = 0.79), the typical drinker (α_T1_ = 0.67, α_T2_ = 0.72), the typical nondrinker (α_T1_ = 0.73, α_T2_ = 0.75), the typical cannabis smoker (α_T1_ = 0.62, α_T2_ = 0.68) and the typical cannabis non-smoker (α_T1_ = 0.72, α_T2_ = 0.77). For this, the participants were asked to imagine an adolescent of their gender and age as a user or non-user of tobacco, alcohol, and cannabis. The participants rated the 6 prototypes across 12 adjectives (smart, confused, popular, immature, cool, confident, independent, careless, unattractive, boring, caring, self-centered) from 1 (not at all) to 7 (extremely). The higher the score for a prototype, the more positively the participant rates the prototype. Explanations of the prototypes and adjectives were presented in the instruction.

The *direct similarity* to the 6 prototypes was measured using a single item for each prototype (e.g., “How similar do you feel (you are like, you look like) to … the typical Tobacco Smoker?”) from 1 (not at all similar) to 7 (extremely similar). The higher the number chosen, the more similar the participant feels to the prototype.

The *indirect similarity* was measured with the difference between self-report (participants rated themselves on the same adjectives as for favorability) and favorability toward each of the prototypes. The lower the difference score obtained, the less different the participant and the prototype were.

The *willingness* to use tobacco (α_T1_ = 0.83, α_T2_ = 0.85), alcohol (α_T1_ = 0.62, α_T2_ = 0.86) or cannabis (*r*α_1_ = 0.65, α_T2_ = 0.64) was measured using situational challenges (e.g., “Suppose you are at a party and several of your friends are smoking. Someone you really like offers you a cigarette”). For each situation, the participants indicated the extent to which they would take and use the substance and the extent to which they would refuse, from 1 (not at all likely) to 7 (very likely). A high score indicates a strong willingness to use the substance.

The *theory of planned behavior* (TPB) [29,30] that was included in the model was measured through our participants’ attitude, perceived norm, perceived control, and intention not to use tobacco, alcohol and cannabis. The participants’ attitude not to use tobacco (α_T1_ = 0.95, α_T2_ = 0.93), alcohol (α_T1_ = 0.90, α_T2_ = 0.94) and cannabis (α_T1_ = 0.96, α_T2_ = 0.97) was measured through 5 adjectives on a 7 points scale (1 = good to 7 = evil, 1 = advantageous to 7 = disadvantageous, 1 = useful to 7 = useless, 1 = pleasant to 7 = unpleasant, 1 = wise to 7 = stupid). A score was obtained with the average of the values chosen for each adjective. The lower the score, the more positive the attitude toward not using. The participants’ norm, control and intention were measured using 2 items each, rated from 1 (I strongly agree) to 7 (I strongly disagree). These three components were used for not using tobacco (α_NORM.T1_ = 0.41, α_CONTROL.T1_ = 0.41, α_INTENTION.T1_ = 0.80, α_NORM.T2_ = 0.85, α_CONTROL.T2_ = 0.54, α_INTENTION.T2_ = 0.83), alcohol (α_NORM.T1_ = 0.85, α_CONTROL.T1_ = 0.45, α_INTENTION.T1_ = 0.88, α_NORM.T2_ = 0.94, α_CONTROL.T2_ = 0.43, α_INTENTION.T2_ = 0.95) and cannabis (α_NORM.T1_ = 0.38, α_CONTROL.T1_ = 0.53, α_INTENTION.T1_ = 0.90, α_NORM.T2_ = 0.84, α_CONTROL.T2_ = 0.18, α_INTENTION.T2_ = 0.93). The lower the score, the more the participants felt that the norm was not to use, that they had control over not using, and that they had no intention of using.

*Tobacco, alcohol, and cannabis use* was measured through the self-reported number of experimenters (those who had used these substances at least once in their lifetime), annual users (those who had used them at least once in the 12 months prior to the study) and recent users (those who had used them at least once in the 30 days prior to the study). They were also asked if they had ever been drunk (yes/no).

#### 2.3.3. Additional Questions

The participants answered various independent questions rated from 1 (not at all) to 7 (completely).

The first two questions focused on the impact of the prevention sessions on their self-perception (“Did your participation in the sessions change your self-perception?”) and their perception of having gained knowledge about themselves (“Did you gain knowledge about yourself?”). The higher the value chosen, the more the adolescents felt that the sessions had changed their perception of themselves and had allowed them to gain knowledge about themselves.

The next questions asked about the impact of the prevention sessions on their perception of having gained knowledge about tobacco, alcohol, cannabis and other drugs (e.g., “Have you gained knowledge about tobacco?”). The higher the value chosen, the more knowledge the adolescents felt that they had gained about each substance. Finally, the students indicated the extent to which they believed the knowledge they had gained would help them change their behavior toward tobacco, alcohol, cannabis and other drugs (e.g., “Will this knowledge help you change your behavior toward tobacco?”). The higher the value chosen, the more likely the adolescents thought that the knowledge gained during the sessions would help them change their behavior toward the various substances.

### 2.4. Statistical Analyses

The SPSS software (version 28) was used for analyses of the scales of the clarity of their self-concept, the favorability toward the prototypes, the direct and indirect similarity to the prototypes, their willingness to use harmful substances, and their attitude, norm, control and intention not to use them. The effect of time (T1/T2), program (Unplugged/IPSELF) and gender (girls/boys) on these scales was measured using repeated measures ANOVA. A sensitivity analysis took into account the middle school effect. Additional questions were measured with an ANOVA test measuring the impact of the program (Unplugged/IPSELF) and gender (girls/boys) on the evolution of self-perception, the acquisition of knowledge about oneself, tobacco, alcohol, cannabis and other drugs, and on the help of this knowledge in changing behavior towards tobacco, alcohol, cannabis and other drugs. The school and student random effects were considered to take into account variabilities between and within schools and students. The results were expressed using means and standard deviations (M, SD), and the value obtained through the F-test and the partial square state (with a small effect size for 0.01 < *η^2^_p_
*< 0.06, a medium effect size for 0.06 < *η^2^_p_
*< 0.14 and a large effect size for *η^2^_p_
*> 0.14).

The Stata software (version 15; StataCorp, College Station, New York, NY, USA) was used to measure the effect of time (T1/T2), program (Unplugged/CTTM), and gender (girls/boys) on the number of experimenters, annual users, and recent users of tobacco, alcohol and cannabis, as well as the extent of drunkenness, using generalized linear mixed models with a logit link function, considering the school and student random effects to take into account variabilities between and within schools and students. The results were expressed using odds ratios (OR) and 95% confidence intervals.

All statistical tests were conducted for a two-sided type I error at 0.05. No correction for multiple testing was applied in the analysis of secondary outcomes or in subgroup analysis. The findings from these analyses were interpreted as exploratory.

## 3. Results

In order to facilitate the readability of this section, the means at T1 and T2 according to program and gender at the different scales are reported in Table 1. The means of girls and boys in the two programs at T1 and T2 are reported in Table 2. The number of consumers was reported in Table 3. A prior analysis (one-way ANOVA) ensured that there was no difference between Unplugged and IPSELF at T1 for the different scales.

### 3.1. Self-Concept Clarity (SCC)

First, we found a clearer self-concept in our participants (*F*(1,150) = 9.412, *p* < 0.01, *η^2^_p_
*= 0.059) between T1 and T2. No difference between Unplugged and IPSELF, either in terms of main effect or interaction with time, was observed. The main effect of gender (*F*(1,150) = 40.495, *p* < 0.001, *η^2^_p_
*= 0.213) indicated that boys (*M* = 4.52, *SD* = 1.18) had a clearer self-concept than girls (*M* = 3.39, *SD* = 1.27). There was no interaction between time and gender.

### 3.2. Prototypes Favorability

Our participants rated the typical drinker more favorably (*F*(1,152) = 4.772, *p* < 0.05, *η^2^_p_
*= 0.030) and the typical nonsmoker (*F*(1,152) = 4.706, *p* < 0.05, *η^2^_p_
*= 0.030), nondrinker (*F*(1,152) = 9.450, *p* < 0.01, *η^2^_p_
*= 0.059) and cannabis nonsmoker (*F*(1,152) = 10.833, *p* = 0.001, *η^2^_p_
*= 0.067) less favorably at T2 than at T1. The main effect of the program indicated that the participants in the IPSELF condition rated the typical nonsmoker (*F*(1,152) = 5.195, *p* < 0.05, *η^2^_p_
*= 0.033, *M_IPSELF_
*= 4.94, *SD* = 0.82, *M_Unplugged_
*= 4.70, *SD* = 0.75) and the cannabis nonsmoker (*F*(1,152) = 4.531, *p* < 0.05, *η^2^_p_
*= 0.029, *M_IPSEL_*_F_ = 4.93, *SD* = 0.78, *M_Unplugged_
*= 4.69, *SD* = 0.71) more positively than those in the Unplugged condition. By contrast, the participants in the IPSELF condition (*M* = 3.63, *SD* = 0.77) rated the typical drinker less favorably (*F*(1,152) = 13.094, *p* < 0.001, *η^2^_p_
*= 0.079) than those in the Unplugged condition (*M* = 3.95, *SD* = 0.69). The interaction between time and program (*F*(1,153) = 4.452, *p* < 0.05, *η^2^_p_
*= 0.028) revealed an increase in favorability toward the typical smoker for participants in the Unplugged condition, versus a decrease for those in the IPSELF condition (see Figure 2). No difference between girls and boys was found, either in terms of main effect or interaction with time.

### 3.3. Prototype Direct Similarity

We observed an increase in direct similarity to the typical smoker (*F*(1,151) = 6.899, *p* = 0.01, *η^2^_p_
*= 0.044) and drinker (*F*(1,151) = 9.404, *p* < 0.01, *η^2^_p_
*= 0.059), as well as a decrease in direct similarity to the typical nondrinker (*F*(1,151) = 5.436, *p* < 0.05, *η^2^_p_
*= 0.035), between T1 and T2. No main effect of the program or interaction with time was found for the different prototypes. However, a main effect of gender (*F*(1,151) = 5.354, *p* < 0.05, *η^2^_p_
*= 0.034) revealed that boys (*M* = 2.58, *SD* = 1.68) felt more similar to the typical drinker than girls (*M* = 2.12, *SD* = 1.39). The interaction between time and gender (*F*(1,151) = 5.614, *p* < 0.05, *η^2^_p_
*= 0.036) also showed us that boys felt more similar to the typical cannabis smoker at T2 than at T1, whereas girls did not experience this difference.

### 3.4. Prototypes Indirect Similarity

The indirect similarity score indicated that at T2 our participants rated themselves as less different from the typical cannabis smoker (*F*(1,152) = 3.942, *p* < 0.05, *η^2^_p_
*= 0.025) and as more different from the typical nondrinker (*F*(1,152) = 10.702, *p* = 0.001, *η^2^_p_
*= 0.066) and cannabis nonsmoker (*F*(1,152) = 7.230, *p* < 0.01, *η^2^_p_
*= 0.045) than at T1. The main effect of the program revealed that participants in the IPSELF condition rated themselves as more different from the typical smoker (*F*(1,153) = 4.464, *p* < 0.05, *η^2^_p_
*= 0.028, *M_IPSELF_
*= 1.44, *SD* = 0.98, *M_Unplugged_
*= 1.08, *SD* = 0.87) and drinker (*F*(1,152) = 14.020, *p* < 0.001, *η^2^_p_
*= 0.084, *M_IPSELF_
*= 1.43, *SD* = 0.95, *M_Unplugged_
*= 0.92, *SD* = 0.85) than those in the Unplugged condition. No interaction effect between time and program was found for the different prototypes. The main effect of gender highlighted that girls (*M* = 0.53, *SD* = 0.49) rated themselves less differently from the typical nonsmoker (*F*(1,152) = 5.258, *p* < 0.05, *η^2^_p_
*= 0.033) than boys (*M* = 0.72, *SD* = 0.84). Moreover, boys (*M* = 0.75, *SD* = 0.71) rated themselves as more different from the typical nondrinker (*F*(1,152) = 5.975, *p* < 0.05, *η^2^_p_
*= 0.038) than girls (*M* = 0.53, *SD* = 0.49). No interaction effect between time and gender was observed for the different prototypes.

### 3.5. Willingness

Our participants’ willingness to use tobacco (*F*(1,153) = 5.553, *p* < 0.05, *η^2^_p_
*= 0.035) and alcohol (*F*(1,153) = 20.152, *p* < 0.001, *η^2^_p_
*= 0.117) increased between T1 and T2. No difference in the effect of program or gender on the willingness to use tobacco, alcohol or cannabis was observed, either in terms of main effect or interaction with time.

### 3.6. Theory of Planned Behavior (TPB)

#### 3.6.1. Attitude

Over time, our participants adopted a less favorable attitude toward not drinking alcohol (*F*(1,153) = 24.308, *p* < 0.001, *η^2^_p_
*= 0.137). No difference related to program or gender, or their interaction with time, was found in attitudes toward tobacco, alcohol or cannabis.

#### 3.6.2. Norm

Our participants’ perceived norm became less favorable toward not using tobacco (*F*(1,153) = 11.426, *p* = 0.001, *η^2^_p_
*= 0.069) and cannabis (*F*(1,153) = 7.069, *p* < 0.01, *η^2^_p_
*= 0.044) over time. No main or interaction effect of the program was observed for the norm of not using tobacco, alcohol or cannabis. A main effect of gender, however, revealed that girls (*M* = 1.14, *SD* = 0.67) perceived a more favorable norm for not using cannabis (*F*(1,153) = 5.505, *p* < 0.05, *η^2^_p_
*= 0.035) than boys (*M* = 1.41, *SD* = 1.14). No interaction effect between gender and time was found.

#### 3.6.3. Control

A loss of perceived control to not use tobacco (*F*(1,153) = 5.620, *p* < 0.05, *η^2^_p_
*= 0.035) and alcohol (*F*(1,153) = 27.504, *p* < 0.001, *η^2^_p_
*= 0.152) was observed between T1 and T2. No main effect of program or gender was found. However, the interaction between time and gender (*F*(1,153) = 4.905, *p* < 0.05, *η^2^_p_
*= 0.031) highlighted a greater loss of control to not use tobacco for girls than for boys. The interaction between time, program and gender (see Figure 3) showed differences in the evolution of the control to not use tobacco (*F*(1,153) = 5.904, *p* < 0.05, *η^2^_p_
*= 0.037), alcohol (*F*(1,153) = 8.060, *p* < 0.01, *η^2^_p_
*= 0.050) and cannabis (*F*(1,153) = 4.605, *p* < 0.05, *η^2^_p_
*= 0.029) according to gender and program. Indeed, for tobacco, we found a gain in control for boys in the Unplugged condition versus a loss of control for girls, while in the IPSELF condition, a loss of control was observed for girls and boys. For alcohol, a greater loss of control was observed for girls than for boys in the Unplugged condition, while on the contrary, it was the boys who had a greater loss of control than the girls in the IPSELF condition. For cannabis, boys in the Unplugged condition had a gain in control and girls a loss of control, whereas in the IPSELF condition, boys had a greater loss of control than girls.

#### 3.6.4. Intention

The intention not to use tobacco (*F*(1,153) = 7.743, *p* < 0.01, *η^2^_p_
*= 0.048), alcohol (*F*(1,153) = 38.389, *p* < 0.001, *η^2^_p_
*= 0.201) and cannabis (*F*(1,153) = 4.247, *p* < 0.05, *η^2^_p_
*= 0.027) decreased between T1 and T2. No effect of program or gender, or interaction with time, was observed.

### 3.7. Substances Use

Due to the small number of cannabis users (experimenters, annual users and recent users), it was not possible to perform a statistical treatment on these data.

#### 3.7.1. Experimenters

A significant increase in the number of tobacco (*OR* = 2.699, *95% CI* [1.252; 5.815], *p* < 0.05) and alcohol (*OR* = 3.750, *95% CI* [1.631; 8.627], *p* < 0.01) experimenters was found between T1 and T2. A main effect of the program (*p* < 0.01) highlighted greater alcohol experimentation among participants in the Unplugged condition (*OR* = 0.409, *95% CI* [0.214; 0.785]). Nevertheless, no interaction between time and program was observed. No gender effect, either in terms of main effect or interaction with time, was found.

#### 3.7.2. Annual Users

An increase in the number of annual tobacco (*OR* = 4.632, *95% CI* [1.335; 16.076], *p* = 0.01) and alcohol (*OR* = 7.615, *95% CI* [2.697; 21.505], *p* < 0.001) users between T1 and T2 was observed. No main effect of program or gender, or interaction with time, was found.

#### 3.7.3. Recent Users

The number of recent tobacco (*OR* = 7.477, *95% CI* [1.300; 43.011], *p* < 0.05) and alcohol (*OR* = 7.271, *95% CI* [1.677; 31.521], *p* < 0.01) users increased between T1 and T2. No difference in the effect of the program or gender, either in terms of main effect or interaction with time, was observed.

#### 3.7.4. Drunkenness

The number of participants who reported being drunk at least once increased between T1 and T2 (*OR* = 21.499, *95% CI* [2.809; 164.559], *p* < 0.01). No main effect of program or gender, or their interaction with time, was found.

### 3.8. Additional Questions

#### 3.8.1. Self

No main or interaction effect of program or gender was observed on the acquisition of self-knowledge.

No program or gender effect was found in the change in self-perception, either in terms of main effect or interaction.

#### 3.8.2. Tobacco

No main effect of the program was found on the acquisition of knowledge about tobacco. Nevertheless, girls (*M* = 5.31, *SD* = 1.52) felt they had gained more knowledge (*F*(1,153) = 5.640, *p* < 0.05, *η^2^_p_
*= 0.036) than boys (*M* = 4.67, *SD =* 1.97). The interaction between program and gender (*F*(1,153) = 7.325, *p* < 0.01, *η^2^_p_
*= 0.046) highlighted that the girls in the IPSELF condition (*M* = 5.61, *SD* = 1.39) estimated gaining more knowledge than the girls in the Unplugged condition (*M* = 5.08, *SD* = 1.58), while for boys, those in the Unplugged condition (*M* = 5.17, *SD* = 1.80) felt that they gained more knowledge than those in the IPSELF condition (*M* = 4.21, *SD* = 2.03) (see Figure 4).

In addition, the participants in the IPSELF condition (*M* = 4.47, *ET* = 2.25) had a stronger sense that acquiring this knowledge would help them change their smoking behavior (*F*(1,153) = 3.900, *p* = 0.05, *η^2^_p_
*= 0.025) than those in the Unplugged condition (*M* = 3.78, *SD* = 2.04). No main effect of gender or interaction between gender and the program was found.

#### 3.8.3. Alcohol

No difference between Unplugged and IPSELF was observed in alcohol knowledge acquisition. As with tobacco, girls (*M* = 5.02, *SD* = 1.77) felt that they had gained more knowledge about alcohol (*F*(1,153) = 4.687, *p* < 0.05, *η^2^_p_
*= 0.030) than boys (*M* = 4.40, *SD* = 1.85). The interaction between program and gender (*F*(1,153) = 5.784, *p* < 0.05, *η^2^_p_
*= 0.036) showed that the girls in the IPSELF condition (*M* = 5.27, *SD* = 1.80) felt that they had gained more knowledge than those in the Unplugged condition (*M* = 4.83, *SD* = 1.73), while the boys in the IPSELF program (*M* = 3.94, *SD* = 1.84) felt that they had gained less knowledge than those in the Unplugged condition (*M* = 4.90, *SD* = 1.77) (see Figure 4).

Nevertheless, neither the program, the gender, nor their interaction influenced a change in behavior related to acquiring this knowledge.

#### 3.8.4. Cannabis

No main effect of program or gender was found in the acquisition of cannabis knowledge. However, their interaction (*F*(1,153) = 7.134, *p* < 0.01, *η^2^_p_
*= 0.045) highlighted that the girls in the IPSELF condition (*M* = 5.76, *SD* = 1.70) felt that they had gained more knowledge about cannabis than those in the Unplugged condition (*M* = 5.47, *SD =* 1.32), whereas the boys in the IPSELF condition (*M* = 4.58, *SD* = 2.00) felt that they had gained less knowledge than those in the Unplugged condition (*M* = 5.67, *SD* = 1.24) (see Figure 4).

No main effect of program, gender or their interaction was observed on a change in cannabis behavior related to the acquisition of this new knowledge.

#### 3.8.5. Other Drugs

No main effect of program or gender was found on knowledge acquisition about other drugs. The interaction between program and gender *(F*(1,153) = 4.461, *p* < 0.05, *η^2^_p_
*= 0.028), as for other substances, indicated that the girls in the IPSELF condition (*M* = 5.63, *SD* = 1.70) felt that they had gained more knowledge about other drugs than the girls in the Unplugged condition (*M* = 5.40, *SD* = 1.34), whereas the boys in the IPSELF condition (*M* = 4.64, *SD* = 1.82) felt that they had gained less knowledge than those in the Unplugged condition (*M* = 5.50, *SD* = 1.59) (see Figure 4).

No main effect of program or gender, or their interaction, showed differences in the extent to which the new knowledge would help change behavior toward other drugs.

## 4. Discussion

This study sought to improve addiction prevention among adolescents through the development of self-concept. This is why we implemented and evaluated the effectiveness of the “Intervention Program based on Self” program by comparing it to the Unplugged reference program. On the one hand, adolescents in the IPSELF program adopted a more favorable image of the typical nonsmoker and cannabis nonsmoker, and a less favorable image of the typical drinker than those in Unplugged. The development of self-concept provided by IPSELF allowed adolescents to feel more different from the typical smoker and drinker over time. More alcohol experimenters were observed among the adolescents in Unplugged. The knowledge gained in IPSELF may also help adolescents more than that gained in Unplugged to change their smoking behavior. On the other hand, IPSELF had a more beneficial effect for girls, who gained more control over their alcohol and cannabis use than boys.

In addition, the girls in the IPSELF program felt that they had gained more knowledge about the substances discussed than those in the Unplugged program.

First, the complementary work on self-concept proposed in IPSELF highlighted differences between it and Unplugged. Adolescents in the IPSELF program rated the typical nonsmoker and cannabis nonsmoker more favorably and the typical drinker less favorably than those in Unplugged. They also came to feel less favorably toward the typical smoker, while those who participated in Unplugged felt more favorably. The literature shows that the more favorable adolescents’ image of the smoker or drinker, the more likely they are to smoke or drink [34,35,36,37,38,39,40]. In addition, the adolescents felt more different from the typical smoker and drinker when they participated in the IPSELF program, and studies have shown that the more similar adolescents’ smoking or drinking images are to their self-images, the greater their intention to smoke or drink [41,42]. In light of the literature on the perception of prototypes, the results observed among adolescents who participated in IPSELF seem to show a more beneficial effect of the program developing self-concept. In fact, we observed a greater number of alcohol experimenters among the adolescents in the Unplugged program. On the other hand, the development of self-concept initiated in IPSELF would help more adolescents to change their smoking behavior with the knowledge gained in the program than would the Unplugged program. Improving one’s self-concept seems to have a beneficial effect on substance use behaviors, as already suggested in the literature [21,22,23]. However, these beneficial effects differ by gender. Indeed, girls in the IPSELF program felt that they had gained more knowledge about tobacco, alcohol, cannabis and other drugs than girls in the Unplugged program, while the opposite was true for boys. Moreover, IPSELF appears to have had a more positive effect on girls’ perceived control over substance use, and Unplugged on boys’ perceived control. Perceived control is an important element predicting behavior [30]. Perceiving a loss of control over substance use increases the risk of their use. Choosing to ignore these gender differences in the effectiveness of the two programs is therefore not an option [7]. It seems preferable to target girls with IPSELF and boys with Unplugged.

Regardless of the differences between the programs, and consistent with the literature on the prototype willingness model in adolescents, we observed on the one hand an increase in favorability toward the typical drinker and a decrease for all typical non-users. In addition, we noted an increase in the feeling of similarity to all typical users and a decrease for the typical nondrinker and cannabis nonsmoker. These elements explain the increase in the willingness to use tobacco and alcohol [36,44]. On the other hand, we found a more favorable attitude towards alcohol use. We also noted a more favorable norm for tobacco and cannabis use. In addition, we found a loss of control over tobacco and alcohol use. These elements explain the increase in the intention to use tobacco, alcohol and cannabis [29,30,31]. These two observations explain the increase in the number of experimenters, annual users and recent users of tobacco and alcohol, and also in the number of adolescents reporting having been drunk at least once [4,26]. We also found an improvement in the clarity of participants’ self-concept, since from adolescence onward this concept undergoes a continuous evolution [43,45,46].

We found the gender differences classically observed in the literature with, first, a clearer self-concept for boys than girls [47]. After puberty, girls are nearly twice as likely to be depressed as boys [48], which may explain the observed gender differences in self-evaluation [49]. Furthermore, due to a tendency toward greater substance use in boys, we observed a greater sense of similarity to the typical smoker and drinker for boys [50]. However, we did not find the gender differences classically observed for substance use [4].

We identified several limitations to our study. The first is the lack of a difference in the evolution of the clarity of one’s self-concept and the feeling of having acquired knowledge about oneself between Unplugged and IPSELF, whereas IPSELF aimed to develop this knowledge. A measure of self-concept positivity might be more appropriate than a measure of self-concept clarity. In addition, work on the self could be incorporated into each session to reinforce its development. Secondly, the low Cronbach’s alpha observed for some two-item scales such as the tobacco and cannabis perceived norm, and tobacco, alcohol and cannabis perceived control, should be noted. The choice of other scales may be considered in the event of a new evaluation of the project.

## 5. Conclusions

In conclusion, in addition to the positive effects of Unplugged already highlighted in the literature, the original project “Intervention Program based on Self”, with its development of self-concept, had a more beneficial effect on user and non-user prototypes perception and on changes in smoking behavior. Moreover, it had a more favorable effect on girls’ perceived control over their use of drugs, whereas Unplugged had a more favorable effect on boys. We therefore recommend the use of IPSELF especially with female audiences.

## Figures and Tables

**Figure 1 ijerph-19-08994-f001:**
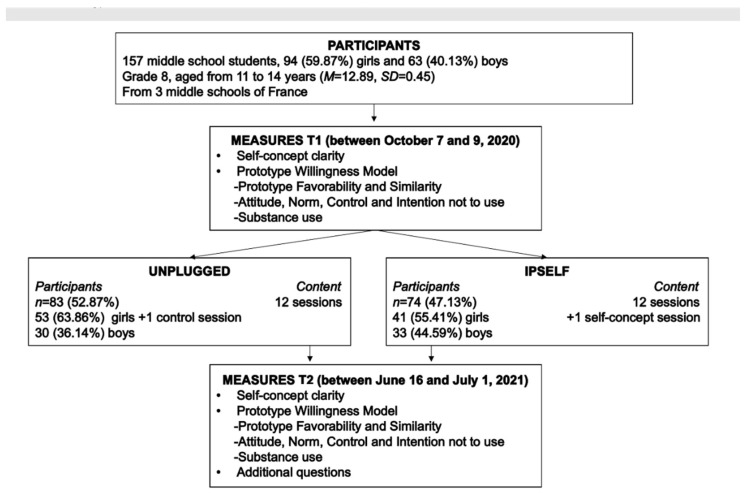
Flowchart of the procedure.

**Figure 2 ijerph-19-08994-f002:**
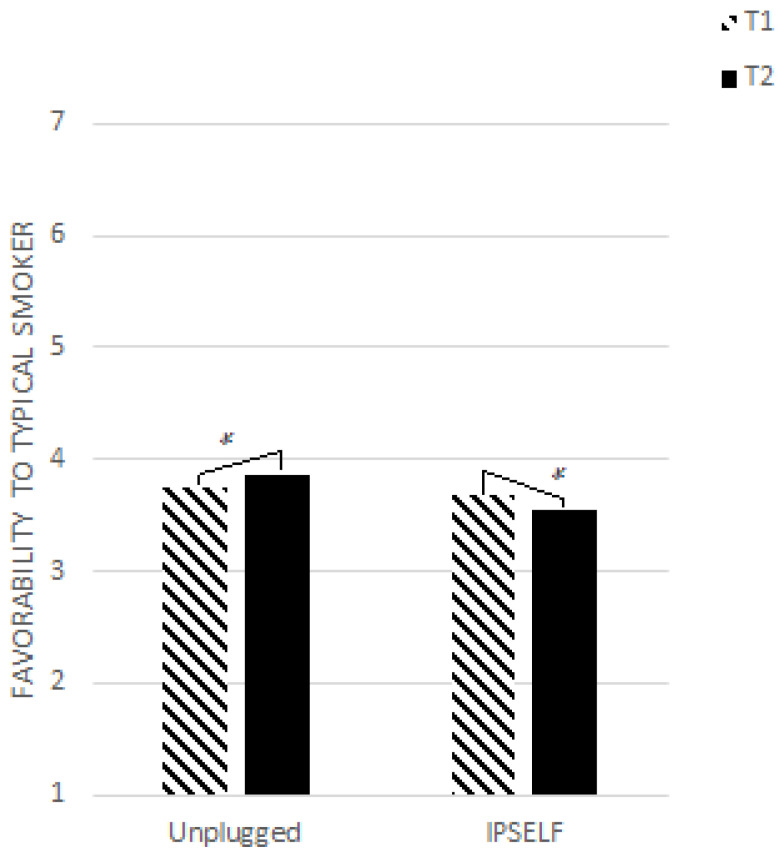
Change in favorability toward a typical smoker by program. * represents the significant effect.

**Figure 3 ijerph-19-08994-f003:**
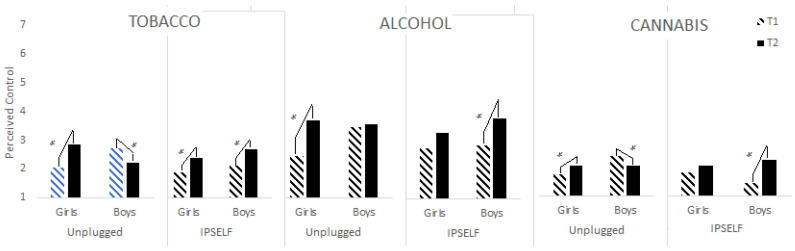
Change in substance use control over time by program and gender. * represents the signifiant effect.

**Figure 4 ijerph-19-08994-f004:**
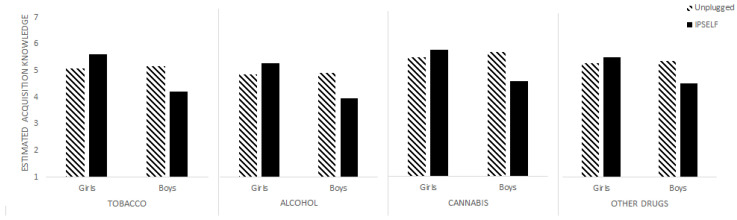
Estimated acquisition of new knowledge about tobacco, alcohol, cannabis and other drugs by program and gender.

**Table 1 ijerph-19-08994-t001:** Means at different scales by program and gender.

		Time 1					Time 2					*Time x Program*
		Total *M(ET)*	Program		Gender		Total *M(ET)*	Program		Gender		
			Unpl. *M(ET)*	*IPSELF* *M(ET)*	Girls *M(ET)*	Boys *M(ET)*		Unpl. *M(ET)*	*IPSELF* * M(ET)*	Girls *M(ET)*	Boys *M(ET)*	
	SCC	3.99(1.35)	3.93(1.37)	4.06(1.34	3.59(1.33)	4.62(1.13)	3.66(1.35)	3.72(1.41)	3.60(1.29)	3.18(1.20)	4.42(1.22)	*p > 0.05*
FAVORABILITY	Smoker	3.72(0.75)	3.75(0.75)	3.68(0.76)	3.69(0.79)	3.75(0.70)	3.72(0.79)	3.86(0.71)	3.55(0.85)	3.62(0.67)	3.87(0.93)	*p* < 0.05 *
	Nonsmoker	4.91(0.78)	4.79(0.81)	5.03(0.74)	4.86(0.73)	4.94(0.87)	4.72(0.80)	4.60(0.69)	4.85(0.89)	4.72(0.73)	4.72(0.89)	*p > 0.05*
	Drinker	3.72(0.74)	3.82(0.68)	3.60(0.78)	3.70(0.71)	3.75(0.78)	3.87(0.75)	4.07(0.69)	3.65(0.76)	3.77(0.72)	4.01(0.78)	*p > 0.05*
	Nondrinker	4.90(0.73)	4.79(0.73)	5.02(0.71)	4.88(0.68)	4.94(0.79)	4.71(0.73)	4.64(0.60)	4.79(0.85)	4.74(0.69)	4.68(0.78)	*p > 0.05*
	Cannabis S.	3.29(0.81)	3.35(0.76)	3.23(0.87)	3.29(0.77)	3.29(0.88)	3.41(0.83)	3.42(0.85)	3.39(0.82)	3.34(0.81)	3.51(0.87)	*p > 0.05*
	Cannabis NS.	4.91(0.75)	4.82(0.78)	5.01(0.70)	4.90(0.72)	4.92(0.79)	4.69(0.76)	4.55(0.64)	4.84(0.85)	4.70(0.74)	4.67(0.78)	*p > 0.05*
SIMILARITY	*Direct*											
	Smoker	1.43(1.15)	1.51(1.20)	1.33(1.08)	1.39(1.16)	1.48(1.13)	1.77(1.43)	1.90(1.46)	1.62(1.37)	1.72(1.37)	1.84(1.53)	*p > 0.05*
	Nonsmoker	5.84(1.66)	5.63(1.70)	6.07(1.59)	5.82(1.61)	5.87(1.76)	5.48(1.70)	5.34(1.64)	5.64(1.77)	5.39(1.72)	5.62(1.69)	*p > 0.05*
	Drinker	2.08(1.42)	2.17(1.50)	1.97(1.33)	1.89(1.28)	2.36(1.58)	2.52(1.62)	2.63(1.55)	2.38(1.59)	2.34(1.49)	2.79(1.77)	*p > 0.05*
	Nondrinker	5.23(1.84)	5.04(1.80)	5.45(1.88)	5.33(1.77)	5.08(1.95)	4.81(1.74)	4.65(1.62)	4.99(1.86)	4.74(1.67)	4.90(1.66)	*p > 0.05*
	Cannabis S.	1.28(0.92)	1.28(0.89)	1.29(0.96)	1.31(1.03)	1.25(0.75)	1.45(1.21)	1.50(1.23)	1.40(1.19)	1.26(0.83)	1.75(1.59)	*p > 0.05*
	Cannabis NS	6.00(1.77)	5.88(1.86)	6.14(1.68)	5.91(1.80)	6.13(1.74)	5.73(1.77)	5.67(1.79)	5.79(1.75)	5.65(1.84)	5.85(1.66)	*p > 0.05*
	*Indirect*											
	Smoker	1.23(0.96)	1.09(0.93)	1.39(0.98)	1.15(0.96)	1.35(0.96)	1.27(0.92)	1.07(0.81)	1.49(0.99)	1.23(0.91)	1.33(0.95)	*p > 0.05*
	Nonsmoker	0.55(0.59)	0.56(0.64)	0.54(0.54)	0.50(0.50)	0.62(0.71)	0.67(0.72)	0.60(0.73)	0.74(0.71)	0.56(0.48)	0.83(0.97)	*p > 0.05*
	Drinker	1.21(1.00)	0.98(0.96)	1.45(0.99)	1.14(0.95)	1.30(1.07)	1.12(0.86)	0.85(0.73)	1.41(0.90)	1.09(0.91)	1.15(0.78)	*p > 0.05*
	Nondrinker	0.54(0.55)	0.51(0.46)	0.56(0.63)	0.49(0.47)	0.61(0.65)	0.69(0.65)	0.60(0.53)	0.79(0.74)	0.57(0.51)	0.88(0.77)	*p > 0.05*
	Cannabis S.	1.61(1.17)	1.40(1.12)	1.84(1.19)	1.49(1.06)	1.79(1.31)	1.47(1.02)	1.35(0.98)	1.61(1.06)	1.44(0.97)	1.53(1.11)	*p > 0.05*
	Cannabis NS	0.58(0.52)	0.59(0.48)	0.57(0.56)	0.52(0.50)	0.67(0.56)	0.71(0.67)	0.63(0.67)	0.80(0.67)	0.63(0.59)	0.82(0.77)	*p > 0.05*
WILLINGNESS	Tobacco	2.39(1.75)	2.60(1.81)	2.17(1.66)	2.30(1.69)	2.54(1.83)	2.79(1.96)	2.96(1.97)	2.59(1.95)	2.79(2.07)	2.78(1.81)	*p > 0.05*
	Alcohol	3.48(1.91)	3.49(1.89)	3.46(1.93)	3.23(1.79)	3.84(2.02)	4.19(1.99)	4.37(1.92)	3.99(2.06)	4.10(2.01)	4.33(1.96)	*p > 0.05*
	Cannabis	1.76(1.50)	1.70(1.44)	1.84(1.58)	1.65(1.36)	1.93(1.69)	1.80(1.43)	1.78(1.42)	1.82(1.45)	1.72(1.45)	1.92(1.41)	*p > 0.05*
TPB	*Tobacco*											
	Attitude	1.97(1.84)	1.93(1.78)	2.02(1.92)	2.11(1.93)	1.76(1.69)	1.84(1.54)	1.67(1.22)	2.02(1.82)	1.90(1.50)	1.74(1.59)	*p > 0.05*
	Norm	1.36(0.88)	1.35(0.77)	1.37(0.97)	1.31(0.79)	1.44(1.00)	1.76(1.46)	1.84(1.52)	1.66(1.40)	1.68(1.36)	1.90(1.61)	*p > 0.05*
	Control	2.15(1.53)	2.30(1.56)	1.99(1.49)	1.92(1.47)	2.41(1.59)	2.57(1.70)	2.61(1.70)	2.52(1.68)	2.64(1.81)	2.46(1.49)	*p > 0.05*
	Intention	1.56(1.23)	1.58(1.16)	1.53(1.31)	1.52(1.19)	1.62(1.28)	1.93(1.57)	2.04(1.64)	1.82(1.49)	1.92(1.55)	1.95(1.63)	*p > 0.05*
	*Alcohol*											
	Attitude	1.76(1.16)	1.76(1.06)	1.96(1.46)	1.82(1.28)	1.64(0.96)	2.40(1.62)	2.22(1.37)	2.60(1.85)	2.32(1.52)	2.51(1.77)	*p > 0.05*
	Norm	1.65(1.17)	1.65(1.18)	1.65(1.17)	1.59(1.14)	1.75(1.22)	1.39(1.25)	1.46(1.41)	1.31(1.01)	1.24(1.01)	1.60(1.53)	*p > 0.05*
	Control	2.84(1.73)	2.86(1.67)	2.82(1.81)	2.61(1.70)	3.18(1.74)	3.65(1.78)	3.72(1.66)	3.57(1.92)	3.58(1.71)	3.75(1.90)	*p > 0.05*
	Intention	2.35(1.72)	2.36(1.76)	2.34(1.69)	2.16(1.59)	2.63(1.87)	3.44(2.11)	3.45(2.04)	3.43(2.21)	3.23(2.07)	3.75(2.15)	*p > 0.05*
	*Cannabis*											
	Attitude	1.48(1.42)	1.49(1.42)	1.46(1.43)	1.40(1.21)	1.59(1.69)	1.74(1.77)	1.66(1.63)	1.82(1.91)	1.58(1.53)	1.97(2.06)	*p > 0.05*
	Norm	1.10(0.54)	1.04(0.33)	1.19(0.70)	1.04(0.33)	1.21(0.74)	1.39(1.25)	1.46(1.42)	1.32(1.04)	1.24(1.01)	1.60(1.53)	*p > 0.05*
	Control	1.87(1.34)	2.05(1.41)	1.66(1.23)	1.82(1.31)	1.94(1.40)	2.13(1.50)	2.11(1.63)	2.16(1.34)	2.09(1.58)	2.20(1.38)	*p > 0.05*
	Intention	1.21(0.88)	1.20(0.88)	1.22(0.88)	1.21(0.85)	1.22(0.94)	1.45(1.27)	1.48(1.44)	1.43(1.06)	1.32(1.09)	1.65(1.49)	*p > 0.05*

* *p* < 0.05, ** *p* < 0.01, *** *p* < 0.001.

**Table 2 ijerph-19-08994-t002:** Means at different scales by interaction between program and gender.

		Time 1				Time 2				*Time x Program x Gender*
		Unplugged		IPSELF		Unplugged		IPSELF		
		Girls *M(SD)*	Boys *M(SD)*	Girls *M(SD)*	Boys *M(SD)*	Girls *M(SD)*	Boys *M(SD)*	Girls *M(SD)*	Boys *M(SD)*	
	SCC	3.59(1.33)	4.57(1.28)	3.60(1.35)	4.66(1.06)	3.28(1.30)	4.56(1.23)	3.04(1.07)	4.31(1.21)	*p > 0.05*
FAVORABILITY	Smoker	3.78(0.73)	3.69(0.79)	3.57(0.86)	3.80(0.60)	3.78(0.45)	4.01(1.00)	3.40(0.83)	3.74(0.86)	*p > 0.05*
	Nonsmoker	4.81(0.72)	4.76(0.97)	4.99(0.74)	5.09(0.74)	4.53(0.46)	4.74(0.97)	4.97(0.93)	4.70(0.83)	*p > 0.05*
	Drinker	3.80(0.62)	3.87(0.79)	3.57(0.80)	3.64(0.77)	3.98(0.56)	4.24(0.87)	3.51(0.82)	3.82(0.64)	*p > 0.05*
	Nondrinker	4.76(0.68)	4.84(0.83)	5.02(0.66)	5.03(0.76)	4.59(0.54)	4.73(0.71)	4.92(0.83)	4.63(0.85)	*p > 0.05*
	Cannabis S.	3.33(0.64)	3.37(0.96)	3.24(0.91)	3.22(0.82)	3.39(0.78)	3.46(1.00)	3.27(0.85)	3.55(0.76)	*p > 0.05*
	Cannabis NS.	4.78(0.73)	4.87(0.87)	5.05(0.69)	4.97(0.72)	4.52(0.55)	4.62(0.78)	4.94(0.89)	4.71(0.79)	*p > 0.05*
SIMILARITY	*Direct*									
	Smoker	1.51(1.23)	1.52(1.15)	1.24(1.04)	1.44(1.13)	2.02(1.62)	1.69(1.17)	1.34(0.83)	1.97(1.81)	*p > 0.05*
	Nonsmoker	5.08(1.55)	5.55(1.98)	6.00(1.67)	6.16(1.51)	5.13(1.68)	5.72(1.53)	5.73(1.73)	5.53(1.83)	*p > 0.05*
	Drinker	1.94(1.32)	2.59(1.72)	1.83(1.24)	2.16(1.44)	2.45(1.53)	2.97(1.57)	2.20(1.45)	2.63(1.95)	*p > 0.05*
	Nondrinker	5.23(1.63)	4.69(2.08)	5.46(1.96)	5.44(1.80)	4.64(1.80)	4.66(1.87)	4.88(1.88)	5.13(1.86)	*p > 0.05*
	Cannabis S.	1.28(0.95)	1.28(0.80)	1.34(1.13)	1.22(0.71)	1.42(1.06)	1.66(1.50)	1.05(0.22)	1.84(1.69)	*p > 0.05*
	Cannabis NS	5.81(1.89)	6.00(1.81)	6.05(1.69)	6.25(1.69)	5.53(1.91)	5.93(1.56)	5.80(1.75)	5.78(1.77)	*p > 0.05*
	*Indirect*									
	Smoker	0.93(0.82)	1.38(1.05)	1.43(1.07)	1.33(0.88)	0.92(0.64)	1.33(1.02)	1.62(1.06)	1.34(0.89)	*p > 0.05*
	Nonsmoker	0.48(0.50)	0.70(0.83)	0.53(0.51)	0.56(0.59)	0.49(0.43)	0.81(1.07)	0.65(0.53)	0.85(0.88)	*p > 0.05*
	Drinker	0.92(0.87)	1.09(1.13)	1.42(0.99)	1.48(1.00)	0.78(0.69)	0.98(0.80)	1.49(1.00)	1.31(0.75)	*p > 0.05*
	Nondrinker	0.46(0.45)	0.60(0.48)	0.53(0.50)	0.61(0.78)	0.51(0.48)	0.77(0.59)	0.65(0.55)	0.97(0.91)	*p > 0.05*
	Cannabis S.	1.27(1.00)	1.64(1.30)	1.77(1.09)	1.92(1.33)	1.25(0.80)	1.53(1.23)	1.68(1.11)	1.53(1.01)	*p > 0.05*
	Cannabis NS	0.54(0.49)	0.68(0.50)	0.49(0.52)	0.66(0.60)	0.57(0.61)	0.75(0.75)	0.72(0.55)	0.89(0.80)	*p > 0.05*
WILLINGNESS	Tobacco	2.55(1.83)	2.68(1.81)	1.98(1.47)	2.41(1.86)	3.21(2.14)	2.53(1.57)	2.26(1.88)	3.00(1.99)	*p > 0.05*
	Alcohol	3.19(1.70)	4.03(2.10)	3.29(1.92)	3.67(1.96)	4.35(1.96)	4.40(1.88)	3.77(2.05)	4.26(2.06)	*p > 0.05*
	Cannabis	1.68(1.46)	1.73(1.42)	1.62(1.23)	2.11(1.91)	1.81(1.57)	1.73(1.15)	1.60(1.29)	2.09(1.61)	*p > 0.05*
TPB	*Tobacco*									
	Attitude	2.01(1.87)	1.79(1.61)	2.24(2.02)	1.73(1.77)	1.81(1.24)	1.42(1.15)	2.02(1.80)	2.02(1.88)	*p > 0.05*
	Norm	1.43(0.90)	1.20(0.45)	1.13(0.60)	1.67(1.26)	1.75(1.41)	1.98(1.71)	1.57(1.29)	1.82(1.54)	*p > 0.05*
	Control	2.05(1.50)	2.73(1.59)	1.89(1.44)	2.12(1.56)	2.84(1.90)	2.22(1.23)	2.39(1.69)	2.68(1.69)	*p < 0.05 **
	Intention	1.64(1.32)	1.48(0.81)	1.37(1.00)	1.74(1.60)	2.19(1.68)	1.77(1.58)	1.57(1.30)	2.12(1.67)	*p > 0.05*
	*Alcohol*									
	Attitude	1.73(1.13)	1.80(0.96)	1.96(1.46)	1.50(0.95)	2.25(1.40)	2.17(1.34)	2.42(1.67)	2.82(2.06)	*p > 0.05*
	Norm	1.57(1.11)	1.80(1.30)	1.61(1.19)	1.70(1.16)	1.43(1.32)	1.50(1.59)	1.00(0.00)	1.70(1.49)	*p > 0.05*
	Control	2.48(1.54)	3.52(1.71)	2.77(1.88)	2.88(1.74)	3.77(1.56)	3.63(1.86)	3.33(1.88)	3.86(1.96)	*p < 0.01 ***
	Intention	2.10(1.51)	2.82(2.07)	2.23(1.70)	2.47(1.68)	3.31(1.97)	3.70(2.17)	3.13(2.22)	3.79(2.17)	*p > 0.05*
	*Cannabis*									
	Attitude	1.45(1.31)	1.57(1.63)	1.34(1.07)	1.62(1.78)	1.71(1.68)	1.57(1.58)	1.41(1.31)	2.32(2.39)	*p > 0.05*
	Norm	1.00(0.00)	1.10(0.55)	1.10(0.49)	1.30(0.88)	1.43(1.32)	1.50(1.59)	1.00(0.00)	1.70(1.49)	*p > 0.05*
	Control	1.81(1.31)	2.48(1.51)	1.83(1.32)	1.45(1.09)	2.11(1.75)	2.12(1.41)	2.06(1.35)	2.27(1.38)	*p < 0.05 **
	Intention	1.15(0.64)	1.30(1.21)	1.28(1.06)	1.15(0.61)	1.42(1.35)	1.58(1.60)	1.20(0.60)	1.71(1.40°	*p > 0.05*

* *p* < 0.05, ** *p* < 0.01, *** *p* < 0.001.

**Table 3 ijerph-19-08994-t003:** Number of experimenters, annual users and recent users of tobacco, alcohol and cannabis by program and gender.

		Time 1					Time 2					*Time Effect*	*Time x Program*
		Total *n(%)*	Program		Gender		Total *n(%)*	Program		Gender			
			*Unpl.* *n(%)*	*IPSELF* *n(%)*	*Girls* *n(%)*	*Boys* *n(%)*		*Unpl.* *n(%)*	*IPSELF* *n(%)*	*Girls* *n(%)*	*Boys* *n(%)*		
TOBACCO	Experimenters	18(11.46)	12(66.67)	6(33.33)	11(61.11)	7(38.89)	43(27.39)	26(60.47)	17(39.53)	22(51.16)	21(48.84)	*p < 0.05 **	*p > 0.05*
	Annual users	12(7.64)	7(58.33)	5(41.67)	7(58.33)	5(41.67)	30(19.11)	17(56.67)	13(43.33)	15(50.00)	15(50.00)	*p = 0.01 ***	*p > 0.05*
	Recent users	4(2.55)	3(75.00)	1(25.00)	3(75.00)	1(25.00)	19(12.10)	12(63.16)	7(36.84)	12(63.16)	7(36.84)	*p < 0.05 **	*p > 0.05*
ALCOHOL	Experimenters	92(58.60)	57(61.96)	35(38.04)	52(56.52)	40(43.48)	122(77.07)	74(60.66)	48(39.34)	72(59.02)	50(40.98)	*p < 0.01 **	*p > 0.05*
	Annual users	61(38.85)	36(59.02)	25(40.98)	33(54.10)	28(45.90)	95(60.51)	57(60.00)	38(40.00)	55(57.89)	40(42.11)	*p < 0.001 ****	*p > 0.05*
	Recent users	13(8.28)	6(46.15)	7(53.85)	5(38.46)	8(34.54)	23(14.65)	16(69.57)	7(30.43)	11(47.83)	12(52.17)	*p < 0.01 ***	*p > 0.05*
	Drunkenness	5(3.18)	4(80.00)	1(20.00)	1(20.00)	4(80.00)	32(20.38)	16(50.00)	16(50.00)	12(37.50)	20(62.50)	*p < 0.01 ***	*p > 0.05*
CANNABIS	Experimenters	2(1.27	1(50.00)	1(50.00)	2(100.00)	0	5(3.18)	2(40.00)	3(60.00)	3(60.00)	2(40.00)	*-*	*-*
	Annual users	1(0.64)	0	1(100.00)	1(100.00)	0	2(1.27)	1(50.00)	1(50.00)	1(50.00)	1(50.00)	*-*	*-*
	Recent users	0	0	0	0	0	0	0	0	0	0	*-*	*-*

* *p* < 0.05, ** *p* < 0.01, *** *p* < 0.001.

## Data Availability

The current study is available in the OSF repository at: https://osf.io/3fq6d/?view_only=4e0760af24f04a87aad67dc4267280e3 (accessed on 25 May 2022).

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
