# Peer review of "“Intervention Program Based on Self”: A Proposal for Improving the Addiction Prevention Program “Unplugged” through Self-Concept"

_ijerph, 2022, doi:10.3390/ijerph19158994_

Round 1
Reviewer 1 Report
In this manuscript, the authors compared two addiction prevention programs to evaluate their effectiveness on a group of middle school students. The program IPSELF differs from the "Unplugged" program on the purpose of developing self-concept in participants, and authors used the Prototype/Willingness Model to compare their outcomes. Teenagers in IPSELF program felt themselves more distant from typical substance users, with less participants that experimented substances during the period of the study, compared to "Unplugged" program. Remarkably, gender differences in the effectiveness of the two programs have been detected, as students experienced more positive effects in perceived control and knowledge acquired on drugs, but girls with IPSELF program and boys with the "Unplugged" program. This study is well-organised and appropriate for this journal. Nevertheless, there are some major and minor issues that need further clarifications.
Major comments:
1. At the end of the Abstract it is fundamental to indicate that "Unplugged" program has more positive effects on boys than girls, as gaining more knowledge and increasing perceived control are more present in male students undergoing this program. This is a pivotal result, as it is the finding that the IPSELF program resulted more appropriate for girls.
2. In the Introduction, the nature of IPSELF program is not completely clear. Is it the "Unplugged" program integrated with a single module designed by the researchers to develop self-concept? Or is it an already used European prevention program, like "Unplugged"? A clarification of this since the introduction is important to avoid confusion in data interpretation.
3. The hypothesis ".. that the knowledge gained about substances during IPSELF will be more helpful in changing teenagers use behavior than those in Unplugged" requires support and deeper explanations, possibily in the context of previous literature supporting such hypothesis. Assertion in line 126 needs a reference too.
4. The terminology Direct similarity and Indirect Similarity should be used not only in the Matherials but in the Introduction section too and their meaning expanded in the interpretation of data regarding prototype favorability.
5. At the end of the Discussion section, limitations to this study should be articulated further, with solutions to the problems indicated or future options to be prospectively considered. Moreover, it is important to reflect that during execution of programs, Unit 10 is the only module that has been conducted virtually on Zoom. This Unit is specifically about acquiring knowledge on drugs, and since data about knowledge on drugs provided interesting results in this study, it would deserve more emphasis and deeper discussion.
Minor comments:
1. Graphical illustration of main results, such as graphs with results for experimenters, perceived control (as an important element predicting behaviour) and favorability to prototypes, would significantly help the reader to appreciate the implications of this study.
2. Line 59: add acronym IPSELF, as it is the first use of the program name "Intervention Program based on Self"
3. Choose if "Unplugged" needs quotation marks or not and use it all the same throughout the paper (be consistent).
4. Line 115: "in the test of the "Intervention Program based on Self", "project" is redundant, please delete.
5. Line 265: "The main effect of gender (F(1,150)=40.495, p<.001, η2 p=0.213) told us that boys (M=4.52, SD=1.18) had a clearer self-concept than girls (M=3.39, SD=1.27)." Similar data interpretations should be limited to the Discussion section.
6. Line 115: "hypothesise", not "hypotheses"; Line 149: "Drugs", not "Drogs"; Line 186: "Cannabis"; Line 240: Correct the meaning of the sentence; Figure 2: "Favorability" on the Y axes; Line 455: "On the other hand", not "in the other hand", the same on Line 469
Author Response
Dear reviewer,
Thank you for taking the time to review our work.
Major comments:
- Thank you for pointing this out. Due to the limited number of words in the abstract section, we had made the choice to present only some of the results. But you are right, it is important. We have added it as requested.
l.27 : “Furthermore, IPSELF had a more beneficial effect for girls, who felt they had lost less control over their alcohol and cannabis use than boys, when Unplugged had a more positive effect on boys, who gained control over their consumption.”
- Thank you for this comment. As explained l.85, IPSELF is an additional self-concept development session to the European Unplugged program
l.85 : “The objective of this study was to implement and evaluate the effectiveness of the "Intervention Program based on Self" program, consisting of the Unplugged program sessions and a complementary session on self-concept development, by comparing it to the reference program "Unplugged" “.
Moreover, in Figure 1, we can see that IPSELF contents 12 Unplugged session + 1 session developing self-concept.
For the sake of clarity, we have completed the sentence explaining the nature of the program
l.62 : “It was in response to this finding that we propose to modify the Unplugged program by adding a specific session; the “Intervention Program based on Self” project (IPSELF) was born. This add-on session to the Unplugged program proposes to bridge this gap with a 13th session developing self-concept.”
l.152 : “Unplugged consists of 12 sessions and IPSELF of 13 (the 12 Unplugged sessions and a session developing self-concept).”
Remember that to avoid any bias, a control session has been added to Unplugged
l.155: “Because the IPSELF program had an additional session, we added a control session for the groups in the Unplugged condition (a game presented as a session working on communication) to avoid a bias related to the number of sessions.”
- This hypothesis was based on the line 64 argument. Nevertheless, we have added theoretical elements to support this hypothesis.
l.64 : “A positive self-concept favors increased effort, perseverance in the face of difficulties, use of acquired abilities and strategies, and increased efficiency [19]. It also allows for better academic performance [20] by facilitating the acquisition of knowledge”.
Justifications have been added to the hypothesis: “we expect no impact of self-concept in attitude, norm or control towards substance use, and thus no impact on intention to use.”, as requested.
l.105 : “This theory has often been used in the prediction of teenagers substance use behavior [31–33], but it does not include the Self, so the development of the self-concept should not act on this dimension of the model.”
- Thank you for pointing this out. In the Prototype model (Gerrard, 2003), the measure proposed is the direct similarity. The participant only indicates how similar he or she feels to the prototype. We have proposed an indirect measure, which is not a direct question to the participants, but corresponds to the difference based the evaluation of the prototype and the evaluation of the participants’ self. As IPSELF proposes a work on the concept of self, it seemed to us to be an added value to integrate this indirect measure, including the self, to the direct similarity.
Indeed, session 10 is the only one to have taken place virtually. However, knowledge acquisition is not the most important determinant of the effectiveness of a prevention program, but rather life skills (Cuijpers, 2002). In addition, this session was the same for both the IPSELF and Unplugged groups (identical content and digital format). Thus, since the change in format was common to both groups, we do not believe that it influenced the results.
As requested, elements have been added to the manuscript
l.162: “This change in format was common to both groups and the content was the same (gaining knowledge about drugs). We know that knowledge acquisition is not the most important element in effective prevention, but rather psychosocial skills. Thus, we do not believe that this change in format will have a significant impact on program effectiveness.”
- Indeed, thank you for this comment. As requested, we have added solutions to the limitations encountered.
l.519 : “A measure of self-concept positivity might be more appropriate than a measure of self-concept clarity. In addition, work on the self could be incorporated into each session to reinforce its development.”
l.524 : “The choice of other scales may be considered in the event of a new evaluation of the project.”.
Minor comments:
- Thank you. Indeed, in order to best respond to this remark, we propose the new Figure 3, illustrating the change in perceived control over time by gender and program for the three substances.
- Thank you for noting that, the add has been made (l.59).
- For the sake of consistency, we have chosen to remove the quotation marks.
- "project" has been deleted, as requested (l.115).
- The significant effect of gender on self-concept (l.287) is discussed line 520.
l.520 : “We found the gender differences classically observed in the literature with first, a clearer self-concept for boys than girls [47]. After puberty, girls are nearly twice as likely to be depressed as boys [48], which may explain the observed gender differences in self-evaluation [49].”
- Thank you for highlighting these various errors that we have corrected.

Reviewer 2 Report
Topic: Substance Use, Disorders and Behavioral Disorders in Primary Care: Prevention, Screening, Care, Support and Coordination
Substance use-related issues have been known worldwide for decades, especially on health and social grounds. According to some recent surveys, it is getting clear that the Covid19 pandemic has been even sharpening such issues, thus highlighting once more the urgency of finding an effective management strategy for substance use behaviours. Furthermore, the role of prevention in this field is becoming more and more evident, as far as psychological and neurobiological processes leading to substance use disorder are highly faceted, and very likely to begin in adolescent age. However, in this regard, scientific literature is still devoid of significative proof concerning deep and stable behavioral changes regarding substance use in young population so far. According to the actual most valuable European prevention programs, prevention must be working on the cognitive approach and on the self-consciousness of adolescent individuals.
Nevertheless, validated high impact methods about prevention still miss in the clinical practice. Moreover, prevention programs should be suitable and feasible for every professional and educational figure among the adolescents’ environment, in order to convey a good prevention practice.
On this topic, the manuscript “Intervention Program Based on Self”: A Proposal Project to Improve the Addiction Prevention Program “Unplugged” through Self-concept” represents an interesting exploration.
The thesis statement is clearly and accurately reported since the title of the manuscript, and the introduction further describes the background idea and the proposal of the authors. The structure of the paper is adequately represented, and the experimental design appears enough consistent and supported by quite adequate references. Conclusions appear to be enough consistent with the evidence and the arguments presented.
Nevertheless, the manuscript seems to present some formal issues which may affect the quality of the presentation, and therefore requiring some important revisions. Namely, some sections may be improved by the following suggestions:
- line 18: the number "3" (referred to schools) might be improve the abstract fluidity if spelled out in "three";
- line 37: since these are the first statements of the paper, the "experiments" here might be better explained if referred to a more specific object (substances? behaviours?), and the following sentence - "preventing addictive behaviours" - should begin with an Effect/Result transition phrase (such as therefore, as a result, thus, etc);
- line 55: "some work has highlighted" should be replaced with "some works have highlighted";
- line 79: a space has to be removed before "This exercise helps..";
- line 92: /where, or whereas/ might fit better in this sentence than "when";
- line 9: a -s is missing at the end of the verb here, as "an individual encounter" is a third singular person;
- line 107: do we need the promo "they" between "believes" and "are"?
- line 112: a comma seems to miss between "self-images" and "the greater";
- line 122; "those" as a pronoun here is referred to?
- line 130: see suggestion at line 18;
- line 149: is it here "Drogs" to be replaced with Drugs?
- line 151: this sentence does not appear too clear. Please consider to replace the verb "full in";
- line 161: "Crombach (alpha)" should be replaced with Cronbach;
- line 179: too many brackets in the sentence (you eventually miss one at the end of the period);
- line 184: the sentence should be concluded by a verb ("was" may be an option here?);
- line 186: a -s has to be deleted from "cannabiss";
- line 218: a space has to be removed before "The higher..";
- line 240-241: a sentence has be left here which it supposed to be previously removed ("the school and...variabilities");
- line 259: a space has to be removed before "A prior analysis..";
- line 263: a space has to be removed between "^2" and "/n/";
- line 282: accordance subject-verb error ("No differences" ... "was found");
- line 296: a comma does not seem to fit between "T2" and "our";
- line 299: a space has to be removed before "The main effect..";
- line 324: a space has to be removed before "No main or..";
- "3.6.3. Control" paragraph: no time accordance among verbs, since in some cases the past simple has been used, in other cases the present simple (see lines 339, 341, 342, for example);
- line 371: an digit error ("thT1")?
- line 373 appears to be in the wrong location (it should be above the table 3, not below);
- line 382: a space has to be removed before "Nevertheless";
- line 391 a space has to be removed before "than those..";
- line 414: something is missing here. Digit error?
- line 430: a space has to be removed between " and Intervention;
- line 436: "knowledges" is a singular noun, it does not need the final -s;
- line 477-479: this sentence has no verb;
- line 490: error digit "develop tit"
- line 491: see line 161
Thank you for your attention.
Author Response
Dear reviewer,
Thank you for your time in reading and reviewing this article. We thank you for your suggestions, which we have taken into account, as indicated below.
- line 122; "those" as a pronoun here is referred to?
“those” refers to teenagers. We used the pronoun to avoid repetition.
- line 151: this sentence does not appear too clear. Please consider to replace the verb "full in";
For more clarity, we have reworded the sentence.
l.156 : “Using the digital format for this session allowed us to finish both programs in all groups”.
- "3.6.3. Control" paragraph: no time accordance among verbs, since in some cases the past simple has been used, in other cases the present simple (see lines 339, 341, 342, for example);
Thank you for noticing. We have corrected the verb tenses in this section.
l.351 : “The interaction between time, program and gender (see Figure 3) showed differences in the evolution of the control to not use tobacco (F(1,153)=5.904, p<.05, η2p=0.037), alco-hol (F(1,153)=8.060, p<.01, η2p=0.050) and cannabis (F(1,153)=4.605, p<.05, η2p=0.029) ac-cording to gender and program. Indeed, for tobacco, we found a gain in control for boys in the Unplugged condition versus a loss in control for girls, while in the IPSELF condition, a loss of control was observed for girls and boys. For alcohol, a greater loss of control was observed for girls than for boys in the Unplugged condition, while on the contrary, it was the boys who had a greater loss of control than the girls in the IP-SELF condition. For cannabis, boys in the Unplugged condition had a gain in control and girls a loss of control, whereas in the IPSELF condition, boys had a greater loss of control than girls.”
- line 414: something is missing here. Digit error?
Indeed a word was missing. We have made the correction.
l.431: “Nevertheless, neither the program, the gender, or their interaction has influenced a change in behavior related to acquiring this knowledge.”
We have also taken into account minor corrections and typos as highlighted which appear in red in the text.

Reviewer 3 Report
I have appreciated vey much this interesting paper. I have only one little suggestion in the Material and method section. You affirm that the two programs was both developed in the 3 schools involved in the reseaerch, schoosing classes by a lottery, but you report only a number of student totally involved. How many classes for each program? How many students for each classes?
Author Response
Dear reviewer,
We thank you for taking the time to review our article and are pleased that you appreciated it.
In order to better respond to your feedback, we have completed certain points in the method section.
l.136 : “Classes in each middle school were randomly assigned (by lottery) to the two pro-grams [Unplugged (n=83 (52.87%) and IPSELF (n=74 (47.13%)], ensuring that each school has the same number of classes for each program.”
l.139 : “Thus, the Unplugged and IPSELF programs were delivered at each middle school to an equivalent number of students to avoid possible differences between middle schools (see Figure 1).”
